# Recent Development and Application of “Nanozyme” Artificial Enzymes—A Review

**DOI:** 10.3390/biomimetics8050446

**Published:** 2023-09-21

**Authors:** Sivakamavalli Jeyachandran, Ramachandran Srinivasan, Thiyagarajan Ramesh, Arumugam Parivallal, Jintae Lee, Ezhaveni Sathiyamoorthi

**Affiliations:** 1Laboratory in Biotechnology & Biosignal Transduction, Department of Orthodontics, Saveetha Dental College and Hospitals, Saveetha Institute of Medical and Technical Sciences (SIMATS), Saveetha University, Chennai 600077, Tamil Nadu, India; 2Centre for Ocean Research (DST-FIST Sponsored Centre), MoES-Earth Science and Technology Cell (Marine Biotechnological Studies), Sathyabama Research Park, Sathyabama Institute of Science and Technology, Chennai 600119, Tamil Nadu, India; srini.vasan70@yahoo.com; 3Department of Basic Medical Sciences, College of Medicine, Prince Sattam bin Abdulaziz University, P.O. Box 173, Al-Kharj 11942, Saudi Arabia; r.thiyagarajan@psau.edu.sa; 4Department of Mathematics, Sungkyunkwan University, Suwon 16419, Republic of Korea; parivallalmaths@gmail.com; 5School of Chemical Engineering, Yeungnam University, Gyeongsan 38541, Republic of Korea

**Keywords:** nanozymes, nanomaterial, biomimetic, enzymatic, catalytic

## Abstract

Nanozymes represent a category of nano-biomaterial artificial enzymes distinguished by their remarkable catalytic potency, stability, cost-effectiveness, biocompatibility, and degradability. These attributes position them as premier biomaterials with extensive applicability across medical, industrial, technological, and biological domains. Following the discovery of ferromagnetic nanoparticles with peroxidase-mimicking capabilities, extensive research endeavors have been dedicated to advancing nanozyme utilization. Their capacity to emulate the functions of natural enzymes has captivated researchers, prompting in-depth investigations into their attributes and potential applications. This exploration has yielded insights and innovations in various areas, including detection mechanisms, biosensing techniques, and device development. Nanozymes exhibit diverse compositions, sizes, and forms, resembling molecular entities such as proteins and tissue-based glucose. Their rapid impact on the body necessitates a comprehensive understanding of their intricate interplay. As each day witnesses the emergence of novel methodologies and technologies, the integration of nanozymes continues to surge, promising enhanced comprehension in the times ahead. This review centers on the expansive deployment and advancement of nanozyme materials, encompassing biomedical, biotechnological, and environmental contexts.

## 1. Introduction

Biomimetics is a term that is used to imply having a duplicate function or mechanism in the field of biological science and technology. Biomimetics are implemented following the principles of various fields, such as biology, physics, architecture, etc., and the biomaterials that are synthesized following these principles possess the functions of the chosen interdisciplinary field, copying and imitating their biological functions [1]. They are made to imitate the principle’s function, be it equipment, machines, systems, or human body parts or tissues.

Biomaterials are materials that are used to make a device or a part of the body, which imitate the biological and physiological functions of the body which are applicable in many fields [2]. Biomaterials need to meet certain guidelines and criteria, such as that they must be biodegradable and biocompatible. They must possess the functions of attachment and cell growth so that the host does not reject the biomaterial through the induction of immunological actions [3]. Biomaterials are classified depending on the origin of the materials, which can be synthetic or natural, and how they are used in medicine (Figure 1).

Synthetic biomaterials are artificial materials used as extracellular microenvironments to mimic the function of, as well as be compatible with, the human body. Some synthetic biomaterials are polymers such as poly lactic-co-glycolic acid, polyester urethane urea (PEUU), peptidomimetic lysine-based poly(ester urethane)urea, chitosan, polyvinylchloride (blood bags), polytetrafluoroethylene (PTFE) (endoscopy, synthetic blood vessels), polyethersulfone (PES) (catheters), carbon fiber (tendons, ligaments, and dental implants), glass fiber (bone cement), poly (methyl methacrylate) (PMMA) (bone cement), polyetheretherketone (PEEK) (dentistry products), ceramics (such as hydroxycarbonate apatite, dicalcium phosphate anhydrous, dicalcium phosphate dihydrate, and tetra calcium phosphate monoxide), metals (316L SS, Ti-based, Co-based, and Mg-based alloys, NiTi, CoCr, and BMG), composites (AI_2_O_3_ and alginate), etc. [4].

Natural biomaterials are biologically derived biomaterials that can be classified into two types, namely non-ECM component mimics, which include cellulose, chitin, chitosan (skin, cartilage, bone, and vascular), alginate, dextran, and silk fibroin; and ECM component mimics such as collagen, gelatin, fibronectin, laminins, elastin, glycosaminoglycan, hyaluronic acid, etc. [5,6,7,8,9,10].

These materials are highly biocompatible and biodegradable, mimicking the functions of drug delivery, antibacterial activity, scaffolds for tissue engineering, tissue regeneration, and a wide variety of functions [11]. Unlike synthetic biomaterials, natural biomaterials are taken from an organism be it plant or animal and are used for repairing the body tissues or to act as a substitute for an organ. They show bioactivity due to their natural extracellular matrix [12]. Natural biomaterials have higher biocompatibility than synthetic biomaterials. Some of the natural biomaterials are collagen, cellulose, alginate, chitosan, etc. [12]. Some of the biomaterials’ applications and functions are dental implants, devices for nerve invigoration, drug delivery, artificial enzymes, protein engineering, mineralization, wound healing, orthopedic applications, endovascular and tissue regeneration, genome technology, and nanotechnology [13].

Nanoenzymes are nano-biomaterial-based synthetic enzymes 1–100 nm in size, with similar structures and functions, such as metal complexes, nucleic acids, glucose, and some other biomolecules. They are classified into two types: (1) enzymes that are changed into nanomaterials called hybrid nanomaterial enzymes, and (2) nanomaterials that imitate enzymatic properties and minimize biocatalytic activity [14]. Since 2007, plenty of studies were carried out on nanozymes after the discovery of ferromagnetic (NPs) nanoparticles with activities like peroxidase. It has been reported that hundreds of nanomaterials possess and mimic the catalytic activity of naturally produced enzymes. Nanozymes are low cost, with high efficiency and high dependability, unlike natural enzymes, which are high cost. The study of nanozymes is called “Nanozymology,” which connects nanotechnology, biology, and science [15].

## 2. Structure and Properties

Even though nanozymes have been an excellent material for implementation based on their low cost, stability, and activity, it has been a difficult task to come up with an ideal design to increase their function and activity. When compared to natural enzymes, nanozymes have a heterogenous surface with a difference in composition and surface structure as they are made of nanomaterials with lower activity. Stating that nanozymes have enzyme-like activity is not accurate. Their function is modified depending on their structure, composition, size, and various other environmental factors [16]. Their activity depends on their structure, which leads to their mechanism being either complex or simple [17]. It is very important to understand and study their structure very precisely so that it can be modified to have higher and more efficient nanozyme activity.

In recent years, there has been a huge number of studies carried out by modifying the nanozyme’s structure to enhance its efficiency. Incorporating and modifying the structure leads to an increase in efficacy of the targeted mechanism. It is important to prioritize the engineering and regulation of the structure and its phytochemical properties [18]. The factors that affect the nanozyme’s activity are its composition, size, structure, position, morphology, valence, and surface modifications, and even other environmental factors based on the applications of such nanozymes [19].

Nanomaterial properties are dependent on size as nanozymes are small and can be exposed more to the active sites of the structure. The catalytic activity is dependent on the interaction with the substrate when it binds [20]. Metal-based nanomaterials such as gold (Au) have higher catalytic activity because they have more interactions with the substrate due to their smaller size, whereas silver (Ag) has a catalytic activity dependent on the pH, the same as Au [20]. A change in pH has a significant effect on the state of H_2_O_2_, and platinum (Pt), being larger, has less catalytic activity. Pt nanozymes 1 nm or smaller show the highest scavenging ability, and those 3 nm in size have increased H_2_O_2_ decomposition [21].

Metal oxide-based nanozyme activity was first reported for Fe_3_O_4_, and further studies on its size and activity were carried out where it was shown that the activity of Fe_3_O_4_ increased when it was smaller in size and with higher POD. CeO_2_ also showed higher SOD-like activity with a smaller size, and considering all of the studies that were carried out, it shows that a smaller nanomaterial leads to higher contact with the substrate and increased catalytic-like activity [22].

Depending on the morphology, the catalytic-like activity differs just like the size factor. Nanozymes show POD-like activity at a low pH and CAT-like activity with a high pH. The decomposition and adsorption depend on the pH. The catalytic-like activity depends on the metal, and in the following order of Pd, Pt, Au, and Ag, the activity rate decreases. The surface activity mechanism is important for many nanozymes, such as iron oxide, etc. [16].

Surface modification is one of the factors that affect the activity of nanozymes with changes in the surface and microenvironment. Surface modifications by small molecules, polymers, covalent, etc. act as a coupling for further attachment of functional groups after acting as a stabilizer for the production and synthesis of nanomaterials. Therefore, it can be adjusted for an increased catalytic effect. For example, Au nanozyme synthesized by using CDs and PAA (sodium polyacrylate) and carbon dots as a soft template, stabilizer and reducing agent, respectively, showed POD-like activity that generated OH, unlike other the activity of other metals or nanomaterials [23].

## 3. Classification of Nanozymes

Nanozymes have been developing alarmingly fast. They can be classified depending on their activity or based on compounds, like iron, selenium, vanadium, carbon, etc., or their catalytic enzyme-like function, such as glutathione peroxidase, oxidases, catalase, superoxide dismutase, hydroperoxide lyase, peroxidase (POD), Au nanoparticles, polypyrrole nanoparticles, and graphene oxide (GO) nanosheets, or the action of the nanozyme biomaterial [24]. Enzymes have the function of catalyzing reactions and are involved in removing toxins, respiratory functions, muscle growth, etc. Nanozymes mimic multiple functions in the human body with a single nanomaterial compound. A compound with an artificial enzyme sometimes plays multi-enzyme roles and functions in the body [25].

Nanozymes also play a major role in technology with the introduction and development of detection devices and equipment with multiple uses in both biology and technology. This has become an area of research interest. These materials contribute a lot to technology, biomedical treatments, detection, and the diagnosis of disease (Table 1). Nanozymes help in sensing and detecting proteins, tumors, molecules, antimicrobial activity, immunoassays, cancer therapy, phenol degradation, etc. [26].

## 4. Biomedical Applications

Research on nanozymes in recent years has made a huge number of developments in sensing, antibacterial activity, cancer treatment, antioxidant activity, and environmental treatment. Nanoparticles with inherent catalytic activity have plenty of biomedical applications, including detection, diagnosis, treatment, and therapy. They are controlled by factors such as hydrogen peroxidase, metal ions, etc. Nanozyme studies proved that many nanomaterials imitate enzyme-like activity, such as oxidase (OXD), glucose oxidase (GOD), peroxidase (POD), catalase (CAT), superoxide dismutase (SOD), and glutathione peroxidase (GPx). The advancement of nanozymes has proceeded to the point where they can be used in the development of advanced variations in different fields.

### 4.1. Antioxidant Activity

Most nanozymes possess antioxidant characteristics, such as eliminating oxidative reactions by eliminating the chain reaction which releases byproduct compounds, causing severe irreversible damage to the human body, such as those occurring in neurodegenerative diseases, renal diseases, tumors, inflammation, Alzheimer’s disease, Parkinson’s disease, etc. ROS cause the death of cells, which affects health. Nanozymes maintain the ROS levels in the body by eliminating excess ROS activity by an antioxidant mechanism.

Nanozymes can act in cytoprotection, where recently, Yang et al., 2023 [31], further studied Pt/Co-SA-NSG. They found that normal mitochondrial function increased with increased antioxidant levels, and the nanozyme decreased inflammation by having superoxide dismutase- and catalase-like activity (Figure 2). Fu et al., 2023 [59], showed and revealed that palladium clusters incubated with insulin (Pd In) reduced the severity of Alzheimer’s disease. Tri-element nanozyme (PtCuSe nanozyme) has potential in treating neurodegenerative disease and Parkinson’s disease by showing catalase and superoxide dismutase activity [60,63].

Two-dimensional cobalt hydroxide oxide nanosheets (Co NSs) with a multienzyme activity of SOD, CAT, and POD protect cells from oxidative damage and scavenge ROS that induce NLRP3 inflammasome and DSS- induced colitis. Co-NSs have increased anti-inflammatory action with therapeutic effects and treatment potential against inflammatory disease [61]. A carbon dot superoxide dismutase (C-dot SOD) nanozyme showed increased potential in healing acute lung injury and protected cells against oxidative damage due to lack of oxygen, and these C-dot SODs scavenge ROS by inhibiting inflammasomes to manage ROS-induced diseases (Figure 3) [62,64].

### 4.2. Cancer Therapy

Nanozymes with catalytic-like activity are used in cancer therapy to induce a hypoxic microenvironment for the tumor cells, limiting the growth of cancer cells, eliminating the cell structure, and inhibiting ROS species. RNA interference is a basic system for gene regulation that is mediated by the RNA-induced silencing complex (RISC). PtFe@Fe_3_O_4_ with dual enzyme activity has a photothermal effect that eliminates tumor cells, causing a hypoxic microenvironment around the tumor cells using peroxidase- and catalase-like activities in the acidic tumor microenvironment (Figure 4) [65].

Shen et al., 2020 [66] carried out research and determined that Ir@MnFe_2_O_4_ NPs have glutathione activity with huge implications for cancer therapy. CoPc-Mn/Ti_3_C2Tx showed highly precise anticancer activity through a multimodal strategy (photoacoustic imaging) for photothermal therapy [67].

Platinum-doped plasmonic gold nanostar–glucose oxidase (Pt-AuNS-GOx) increases the efficacy of hypoxia alleviation and cancer treatment, induces an energy barrier reduction for oxygen production in acidic and aerobic environments, inhibits the hypoxic microenvironment of tumor tissues, increases apoptosis, and inhibits metastasis of cancer with therapeutic efficacy [68].

### 4.3. Antibacterial Activity

Millions of people each year have been affected by infectious illnesses caused by bacteria up to this point, ranking as one of the largest global health issues. Recently, Shi et al., 2022 [69] designed porous graphitic carbon nitride C_3_N_5_ nanosheets (denoted as PtRu/C_3_N_5_) that are piezo-augmented and photocatalytic nanozyme-integrated microneedles, which imitate oxidase activity and are nanotherapeutics with multiple activities. They highlighted their use in therapeutic strategies based on their antibacterial and anti-inflammatory actions. NiCo_2_O_4_ exhibits a high antibacterial effect without any worry about antimicrobial resistance, imitating the biological mechanism of antibodies [70]. N-CNDs inhibit bacterial growth by exhibiting antibacterial action against both Gram-positive and Gram-negative bacteria, as carbon-based nanozymes have multiple activities and are used in food safety from bacterial contamination [71]. Most of the helper cells are seriously changed as the tributer norms are reached.

### 4.4. Neurodegenerative Disease Therapy

Therapies aimed at neurodegenerative diseases necessitate the precise regulation of intercellular levels of reactive oxygen species (ROS), given their close association with the pathogenesis of nervous system disorders. Consequently, nanozymes exhibiting ROS scavenging capabilities hold immense potential for effective neuroprotection. Drawing inspiration from the pivotal role of manganese (Mn) in the catalytic processes of natural superoxide dismutase (SOD) enzymes, Mn-based nanozymes have garnered significant attention as antioxidants. In pursuit of nanozymes with potent and broad-spectrum antioxidant prowess, Singh et al. engineered the morphology of Mn_3_O_4_ nanoparticles (NPs) and introduced a biocompatible flower-like Mn_3_O_4_ nanostructure (referred to as Mnf), which efficiently combats elevated ROS levels under pathophysiological conditions. The Mnf exhibits multiple enzyme-like activities, encompassing SOD-, catalase (CAT)-, and glutathione peroxidase (GPx)-like functions, surpassing those of Mn_3_O_4_ NPs with alternative morphologies. In vitro studies substantiate the robust cytoprotective effects of Mnf against neurotoxin-induced cell death in SHSY-5Y cells, thereby holding promise for mitigating ROS-mediated neurodegenerative diseases like Parkinson’s disease. Furthermore, it has been reported that cerium nanoparticles (CeNPs) smaller than 5 nm demonstrate exceptional catalytic properties akin to CAT and SOD [71]. Building upon this discovery, our research group engineered a nanocomposite loaded with methylene blue (MB) and adorned with CeNPs on its surface, presenting a notable therapeutic avenue for Alzheimer’s disease (AD) treatment. The CeNPs effectively scavenge intracellular ROS, mitigating mitochondrial oxidative stress and curtailing tau hyperphosphorylation when combined with MB. This comprehensive approach alleviates AD symptoms both in vitro and in vivo, underscoring the potential of addressing mitochondrial dysfunction in neuroinflammation for the treatment of various neurodegenerative diseases.

### 4.5. Injury Therapy

Excessive levels of reactive oxygen species (ROS) and inflammation represent significant challenges that hinder the recovery and treatment of injuries. This is because the resulting oxidative stress triggers the opening of inter-endothelial junctions and facilitates the migration of inflammatory cells across the endothelial barrier. Hence, it becomes imperative to counteract the surplus ROS and promote oxygen (O_2_) generation within damaged cells and injury sites. This can be achieved through the utilization of nanozymes endowed with superoxide dismutase (SOD)- and catalase (CAT)-like activities. For instance, Zhang et al. devised micelle-like nanoparticles by conjugating polyethylene glycol (PEG) with manganese protoporphyrin (PEG-MnPP) for the treatment of acute liver failure. These PEG-MnPP nanoparticles, with extended circulation in the bloodstream, exhibit CAT- and SOD-like activities, effectively neutralizing hydrogen peroxide (H_2_O_2_) for enhanced management of acetaminophen (APAP)-induced acute liver failure. Singh et al. engineered cerium nanoparticles (CeNPs) with CAT-like activity, successfully safeguarding human hepatic cells from a catalasemia induced by 3-AT. Moreover, our research group developed PEG-coated CeNPs with CAT- and SOD-like activities, offering a highly efficient approach for treating drug-induced liver injury (DILI). CeNPs can directly scavenge ROS, detoxifying DILI. Additionally, the O_2_ produced during CeNPs’ ROS-scavenging process effectively inhibits pro-inflammatory macrophages, mitigating inflammation-induced damage to liver tissue. Consequently, the dual roles of CeNPs in detoxification and inflammation regulation significantly extend the therapeutic window for DILI treatment compared to conventional N-acetylcysteine.

Furthermore, Huang et al. harnessed citric acid-modified CeNPs to alleviate rhabdomyolysis-induced acute kidney injury by providing robust protection to renal cells against ROS. In a separate study, Mugesh and colleagues demonstrated that cerium vanadate (CeVO4) nanorods exhibit SOD-like activity within cells, even when the natural enzyme is down-regulated due to specific gene silencing. CeVO4 effectively shielded SOD-depleted cells from mitochondrial damage by simultaneously regulating superoxide levels and restoring the function of antiapoptotic Bcl-2 family proteins. This preservation of mitochondrial integrity enabled the efficient regulation of ATP levels in neuronal cells under oxidative stress, showcasing CeVO4′s immense potential for treating diseases associated with mitochondrial dysfunction. Overall, chemically designed nanozymes with antioxidant activities have shown remarkable efficacy in managing diseases related to oxidative damage, particularly in the context of injury therapy. Moreover, these nanozymes can compensate for functional deficits resulting from genetic defects, underscoring their versatile therapeutic potential.

### 4.6. Biosensor

Biosensors have been used for hundreds of years for the detection of various compounds, such as molecules, ions, metals, proteins, cancer cells, nuclei acids, pesticides, etc. Many biosensor strategies have been built upon by researchers, such as in the form of electrochemical biosensors, colorimetric sensors, surface-enhanced Raman spectroscopy (SERS), and florescent meters. These were developed after clear knowledge was achieved on nanomaterials as well as metal oxides and frameworks [72]. Natural biomaterials are biologically derived biomaterials and are classified into two types, non-ECM component mimics, such as cellulose, chitin, chitosan (skin, cartilage, bone, and vascular tissue), alginate, dextran, and silk fibroin, and ECM component mimics, such as collagen, gelatin, fibronectin, laminins, elastin, glycosaminoglycan, hyaluronic acid, etc. [5].

Electrochemical biosensor-integrated devices are widely used in the medical field for the detection of breast cancer [73], infectious viral disease [74], food-borne pathogens [75], glucose and pH [76], etc. The researchers built upon new and recent findings on incorporating nanomaterials and developing methods and devices. Ag-Fe_3_O_4_ nanozymes were developed with a peroxidase-like activity for the identification of sulfur ions colorimetrically. They have a dual activity, where they catalyze the degradation of triarylmethane dye at a rate of 99%, and can be reused ten times or more [77]. Single-atom nanozymes (SANs) are used as electrochemical sensors due to their stability, selectivity, and increased sensitivity to detect molecules such as hydrogen peroxide, glucose, oxygen, and uric acid for monitoring food safety and human health. It is accurate and has high quantitative sensitivity to eliminate the negative impacts on food safety [78].

## 5. Nanozymes in Biodiversity and Environmental Treatment

With the development of industries, there is increased environmental pollution caused by humans, which disturbs ecological systems and disrupts the life cycle of human beings. The pollution caused by humans, such as the disposal of wastes such as oil, drainage, heavy metals, and other waste into water, is an increasing problem. By applying bio-technology, the factors that contribute to pollution can be solved, and treatment can be strategized. Catalase, superoxide dismutase, and other peroxidase activities are often applied in the remediation of the environment (Figure 5) [63,79].

Nanozymes can sense fertilizers, heavy metals, pesticides, and other biological factors, which can be used in implementing treatments for pollutant degradation, and bio-fouling processes. The detection of molecules and organic compounds quantitatively and qualitatively can be carried out with nanozymes. They can degrade dyes, phenols, persistent pollutants, toxic pollutants, and chemical agents [80]. Portable and recyclable agarose gel films loaded with CeO_2_ @ZIF-8 nanozymes were used to study the degradation of dye in wastewater and showed an 80% efficacy after 5 uses. A non-photodegradation catalytic system with even better development can be implemented for environmental remediation and eliminating dye pollution [81]. Co_3_O_4_/CoFe_2_O_4_ hollow nanocubes (HNCs) have a multifunction and multienzyme activity for the detection of L-cysteine in food and degradation of rhodamine B by 99.24%, and can be reused more than 20 times for the detection of norfloxacin in drugs [9,82]. CH-Cu nanozymes were reported to have a higher efficiency than laccase in the degradation of phenolic pollutants, such as chlorophenols and bisphenols, and based on these nanozymes, a detection method for epinephrine was developed where a smartphone can perform the quantitative analysis [83,84].

## 6. Conclusions

These new-generation nanozymes with high catalytic activity and low cost make excellent materials for carrying out further studies and development. They exhibit a great capability for disease therapy, detection, biosensing etc. In this review, we have briefly summarized the recent developments and findings on nanozyme functions and applications. Nanozymes with enzyme-like activity can be utilized tremendously to our advantage. Nanozymes with high biocompatibility and stability received attention and made it possible to develop and apply nanomaterials for sensing, antibacterial activity, antioxidant activity, and tumor therapy. Most nanozymes showing dual enzymatic activity and functions can bring greater results. The selectivity of these nanozymes for development can be further studied, and research can be carried out. With the future in consideration, a lot of developments and implementations can be made with nanozymes in the biomedical field.

## Figures and Tables

**Figure 1 biomimetics-08-00446-f001:**
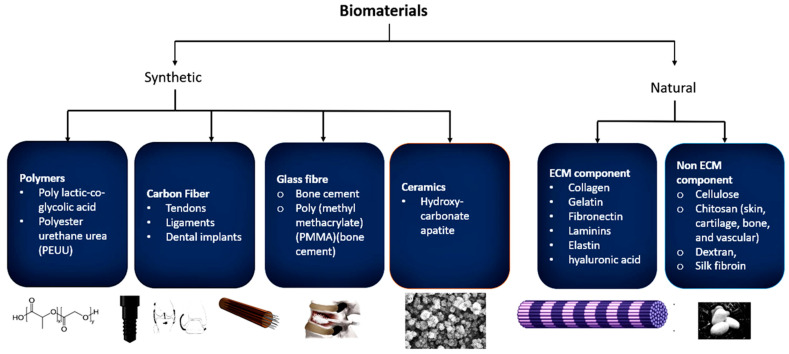
General classification of biomaterial types and some examples: synthetic and natural.

**Figure 2 biomimetics-08-00446-f002:**
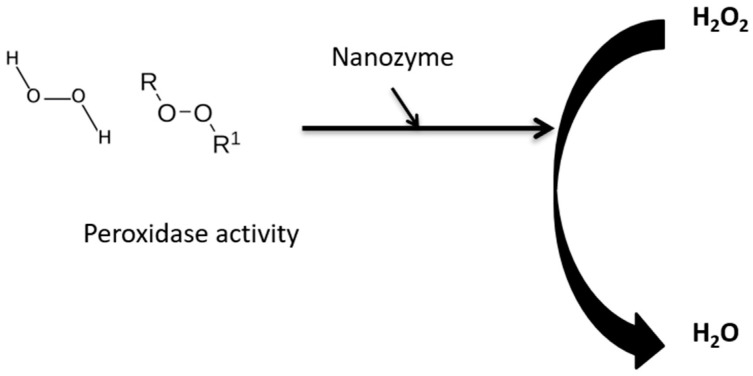
Role of nanozymes in peroxidase catalytic activity.

**Figure 3 biomimetics-08-00446-f003:**
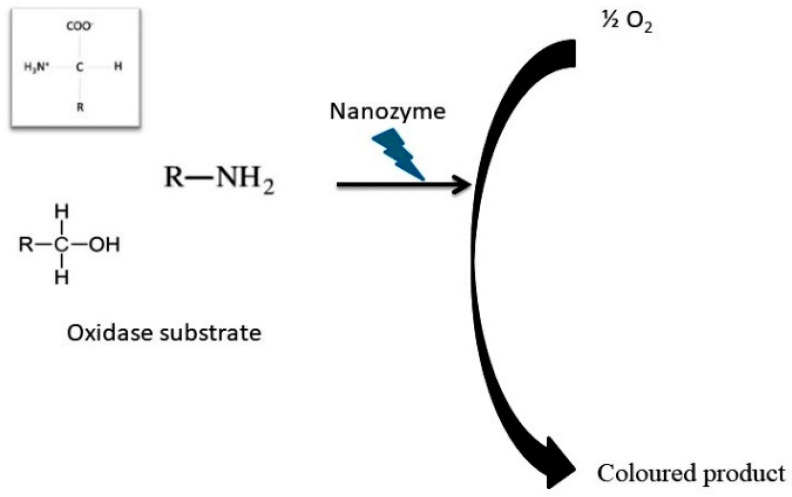
Role of nanozymes in oxidation catalytic activity.

**Figure 4 biomimetics-08-00446-f004:**
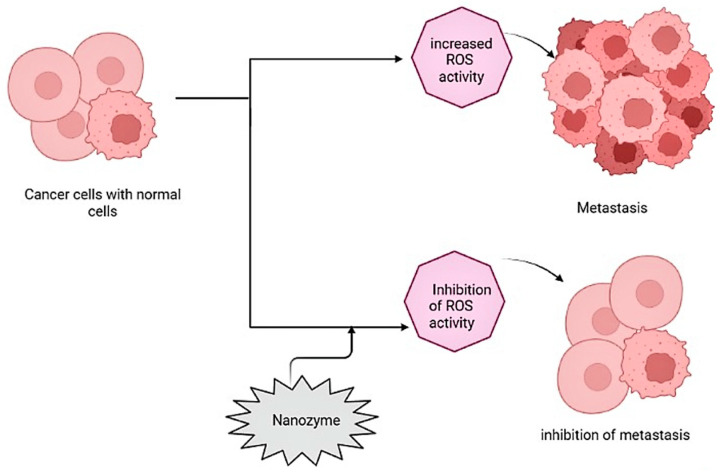
Illustration of antitumor mechanism/activity by nanozymes against cancer cells.

**Figure 5 biomimetics-08-00446-f005:**
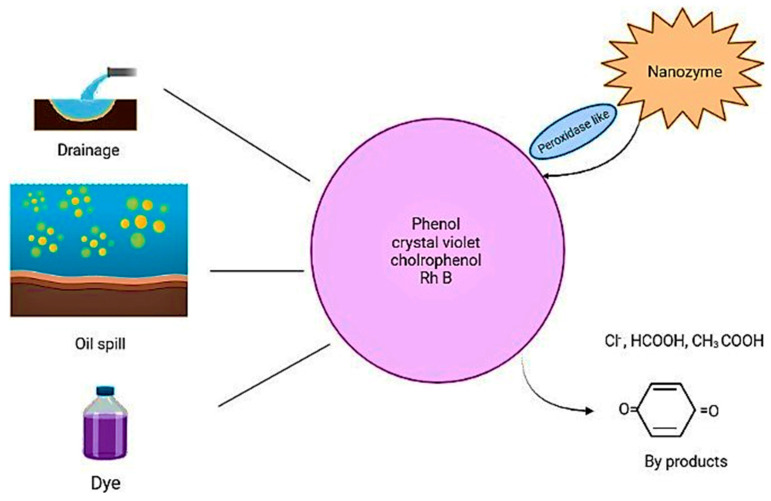
Mechanisms behind the use of nanozymes in degrading environmental pollutants.

**Table 1 biomimetics-08-00446-t001:** Nanomaterials and their biomedical applications with enzymatic activities.

Metal/Compound— Nanomaterial	Biomimetic Enzyme Activity	Biomedical Application	Reference
Vanadium (V) o-PDAox Vanadium vacancy-rich BiVO_4_ nanosheets (Vv-riVO_4_ NSs) Vanadium nitride mxene (V_2_N) nanosheets Vanadium-incorporated dendritic mesoporous silica (VMSN) VC@Lipo Vanadium–iron Prussian blue analog impregnated with insulin (V1/5Fe(CN)_6_PBA/INS) 2D nanosheet-like V_2_O_5_ (2D-VONz)	Photo-responsive oxidase, glutathione peroxidase, catalase, superoxide dismutase, hydroperoxide lyase, POD	Colorimetric biosensor for the detection of L-arginine Mitigates tumor cell growth by improving piezoelectric therapy Prevents normal tissue from being damaged by hyperthermia Therapeutic effects that terminate bacterial growth and improve healing Identified dopamine (DA) development using colorimetric sensing Increased cytotoxicity causing harm for SH-SY5Y neuroblastoma cells and treated cancer cells Inspection of the rate of fish freshness A portable device that is paper-based for the detection and sensing of pesticides directly	[27,28,29,30,31,32]
Gold (Au) Liposomes @gold nanoparticle– tirapazamine hybrid (Lip@Au-TPZ) Gold nanozymes (GNZs) Ti_3_C_2_Tx MXene nanoribbons@gold (Ti_3_C_2_Tx MNR@Au)	Glutathione peroxidase, catalase, phosphatase, superoxide dismutase, POD, helicase	Elimination of biofilm caused by bacteria with enhanced hypoxia Damage of bacterial DNA by TPZ Enhances or inhibits the catalytic action Modification of surface enabling bioassays and biosensing Detection of heavy metals and sensing the presence of mercury volumetrically	[33,34,35,36]
Platinum (Pt) Pt/Co-SA-NSG ICPA Mesoporous platinum nanozymes (MPNs)	Glutathione peroxidase, catalase, polyphenol oxidase (PPO), superoxide dismutase, lipogenesis, POD	Eliminating unnecessary ROS and treating ear irritation brought on by ROS Improves mitochondrial function and in turn, the ATP levels are regulated Increased antioxidant activity reduces factors that cause inflammation Accumulates in tumor tissue and causes hypoxic microenvironment in a self-strengthening manner Is an immunoassay for the detection of 25(OH)VD3 by sensitive colorimetrics	[37,38,39,40]
Selenium (Se) Selenium-enriched Prussian blue nanozymes (Se-HMPB nanozymes) GOx@IM-Se-Ph@TA-Fe NPs PDANPs DSeP@PB	Glutathione peroxidase, catalase/POD/superoxide dismutase/OXD	Inhibits ferroptosis and protects the intestinal barrier in ulcerative colitis by reversing the peroxidation of epithelial cells Potential in decreasing intestinal bowel disease Efficient chemodynamic therapy and increased antioxidant activity Stimulates angiogenesis and inhibits inflammation, which increases wound healing Antibacterial activity, immunomodulator, and a ROS scavenger	[41,42,43]
Copper (Cu) CuNAD Cus-QCS CP@CA CD47	Glutathione peroxidase, catalase, superoxide dismutase, POD	Chromogenic detection of cholesterol Electrostatic bonding damages the membranes and eliminates the bacteria, improving wound healing Revitalize the environment by degrading sulfonylurea herbicides in soil and the environment Treatment of breast cancer by inhibiting the phagocytic activity of macrophages Chemodynamic and photothermal tumor treatment	[44,45,46]
Ruthenium (Ru) Polyvinylrrolidone-stabilized platinum ruthenium nanozyme (PVP/PtRu NZs)	Glutathione peroxidase, catalase, superoxide dismutase, POD, photo-responsive oxidoreductase	Estimation of lysophosphatidylcholine concentration in serum Protection against ROS damage and cures acute renal injury	[47,48]
Manganese (Mn) Mn-UMOF Polydopamine—manganese-organic framework (pda-MNOF) MnSA−N_4_−C MnO_2_-NEs	Glutathione peroxidase, catalase, superoxide dismutase, POD, oxidase	Sensing of L-cysteine by developing a fluorescent turn-off sensor Protects the cells from injury caused by excess ROS activity Induces vascular cellular growth (endothelial cells) An agent for stroke treatment with increased antioxidant activity Colorimetric biosensing platforms Detection of biomolecules Nanocarrier for drug delivery, sonosensitizers, and photosensitizer Improved clinical imaging diagnostics	[49,50,51,52]
Iron (Fe) Safs@LGOX Iron-site-containing poly-γ-glutamic acid/chitosan hydrogel nanoparticle (PGA-Fe/CS NP) IONPs P450-like enzymatic activity	Oxygen reduction reaction (ORR), glutathione peroxidase, POD, catalase, superoxide dismutase	Reactor for α-ketoglutarate synthesis A microfluidic device made of foldable paper was developed to identify target biomolecules Reduces the risk of immunogenicity Increased inhibition of tumor proliferation and acts as tumor therapy by drug loading with a dual activity	[53,54,55,56]
Cerium (Ce) Ce_3_+ Dextran-coated cerium oxide (D- CeO_2_) MoS_2_-bPEI-CeFe_2_O_4_ NFs	Glutathione peroxidase, catalase, superoxide dismutase, POD	Decreases bone loss and fracture caused by ionizing radiation Protect cells from oxidative damage Induces oxidation reactions and scavenges ROS Imaging of gastrointestinal tract by CT contrast agent Therapeutic effects for cancer by eliminating cancer cells	[57,58,59]
Palladium (Pd) GeO_2_@Pd–H_2_O_2_-TMB A-Pd@MoO_3_–x NH MIL-88@Pd/Pt	Catalase, super oxide dismutase, POD, Oxidase	Detection of pesticides in samples (water) Induces synergistically enhanced cascade catalysis for tumor-specific therapy Sensitive colorimetric biosensing of pathogens	[60,61,62]

## Data Availability

Not applicable.

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
