# Peer review of "Recent Development and Application of “Nanozyme” Artificial Enzymes—A Review"

_biomimetics, 2023, doi:10.3390/biomimetics8050446_

Round 1
Reviewer 1 Report
1- Language should be carefully revised throughout the manuscript.
2- Many types of research concerned with the application of Nanozymes in Environmental fields should be handled in the offered manuscript.
3- The number of references is not appropriate with the huge number of reported studies that are concerned with Nanozymes. So, the literature reviews should be enlarged and carefully written.
Language should be carefully revised throughout the manuscript.
Author Response
1- Language should be carefully revised throughout the manuscript.
Ans: The English language has been revised throughout the manuscript.
2- Many types of research concerned with the application of Nanozymes in Environmental fields should be handled in the offered manuscript.
Ans: Some more applications of Nanozymes in other sectors have been included.
3- The number of references is not appropriate with the huge number of reported studies that are concerned with Nanozymes. So, the literature reviews should be enlarged and carefully written.
Ans: Additional references has been added as per the reviewer suggestion.

Reviewer 2 Report
The review "Recent development and application of “Nanozymes” artificial enzymes" is timely and significant manuscript. The use of nanozymes in biomedical application is a rapidly developing area and new reviews on this topic will be useful to the research community. However, I recommend to accepted this manuscript after significant changes
Some issues for the authors to consider are these;
1) There are no references to figures in the manuscript text
2) Fig 1. Classification of biomaterials. It's not clear for me what the authors wanted to demonstrate at the bottom of the figure (3rd image from the left).
3) Introduction. Line 51-61. The authors mentioned synthetic biomaterials and their applications, listing at least 10 different compounds. But there is only one reference to a chapter in a book at the end of a paragraph. For the review, it is desirable to provide links to original studies that show the actual use of these materials.
4) 2. Structure and properties. Line 104-105. The authors mention that incorporating and modifying the nanozymes structure leads to an increase in efficacy, but there is also no links to original studies that demonstrated this
5) 4.1. Antioxidant activity. Line 171-174. The sentence need to be rewritten.
6) In my opinion, this review does not adequately describe the difficulties in working with nanozymes, which must be resolved before translating these materials into practice.
Author Response
The review "Recent development and application of “Nanozymes” artificial enzymes" is timely and significant manuscript. The use of nanozymes in biomedical application is a rapidly developing area and new reviews on this topic will be useful to the research community. However, I recommend to accepted this manuscript after significant changes
Some issues for the authors to consider are these;
1) There are no references to figures in the manuscript text
Ans: References to figures has been added
2) Fig 1. Classification of biomaterials. It's not clear for me what the authors wanted to demonstrate at the bottom of the figure (3rd image from the left).
Ans: The query in the figure has been sorted.
3) Introduction. Line 51-61. The authors mentioned synthetic biomaterials and their applications, listing at least 10 different compounds. But there is only one reference to a chapter in a book at the end of a paragraph. For the review, it is desirable to provide links to original studies that show the actual use of these materials.
Ans: suitable has been added as per reviewer suggestion.
4) 2. Structure and properties. Line 104-105. The authors mention that incorporating and modifying the nanozymes structure leads to an increase in efficacy, but there is also no links to original studies that demonstrated this
Ans: Original research papers has been referred and included citations.
5) 4.1. Antioxidant activity. Line 171-174. The sentence need to be rewritten.
Ans: Rephrased
6) In my opinion, this review does not adequately describe the difficulties in working with nanozymes, which must be resolved before translating these materials into practice.
Ans: Changes made for the reviewer suggestion

Reviewer 3 Report
In the paper entitled “Recent development and application of “Nanozymes” artificial enzymes - a review” Jeyachandran et al. highlight on “nanozyme”; their structure-functions and applications. Firstly, they classified biomaterials as per natural and synthetic-followed by structural diversity and biomedical applications. This topic has indeed broad implications in modern day research and thus a review will lead to further development in future.
In section 2 the authors have emphasized on different type of nanozymes but consisting of nanomaterials only. It would be better for the sake of this review if they could provide some recent literature on functionalized nanomaterials (“Functionalized gold nanomaterials as biomimetic nanozymes and biosensing actuators”, “Functional catalytic nanoparticles (nanozymes) for sensing” and so on)-this will probe the role of the ligands that enhance catalytic properties of nanomaterials. Similarly, role of short peptides needs to be addressed for betterment of this review (“Short Peptides in Minimalistic Biocatalyst Design”, “Peptide-Gold Nanoparticle Conjugates as Sequential Cascade Catalysts”, “Molecular Dynamics Simulations of a Catalytic Multivalent Peptide–Nanoparticle Complex” etc.).
A short section on existing methods (both experiments and theory) will also increase impact of the review.
Thus, after incorporating the necessary changes the review will be more appealing to the readers and it will be worth publishing in “Biomimetics”.
Author Response
In the paper entitled “Recent development and application of “Nanozymes” artificial enzymes - a review” Jeyachandran et al. highlight on “nanozyme”; their structure-functions and applications. Firstly, they classified biomaterials as per natural and synthetic-followed by structural diversity and biomedical applications. This topic has indeed broad implications in modern day research and thus a review will lead to further development in future.
In section 2 the authors have emphasized on different type of nanozymes but consisting of nanomaterials only. It would be better for the sake of this review if they could provide some recent literature on functionalized nanomaterials (“Functionalized gold nanomaterials as biomimetic nanozymes and biosensing actuators”, “Functional catalytic nanoparticles (nanozymes) for sensing” and so on)-this will probe the role of the ligands that enhance catalytic properties of nanomaterials. Similarly, role of short peptides needs to be addressed for betterment of this review (“Short Peptides in Minimalistic Biocatalyst Design”, “Peptide-Gold Nanoparticle Conjugates as Sequential Cascade Catalysts”, “Molecular Dynamics Simulations of a Catalytic Multivalent Peptide–Nanoparticle Complex” etc.).
Ans: A brief part has been added regarding the functionalized nanomaterials and short peptides has been added.
A short section on existing methods (both experiments and theory) will also increase impact of the review.
Ans: As per the reviewer suggestion changes have been made.
Thus, after incorporating the necessary changes the review will be more appealing to the readers and it will be worth publishing in “Biomimetics”.
Reviewer 4 Report
The submitted manuscript makes a brief review with the topic of “nanozyme”, one emerging field of the research frontier. Based on a careful consideration, it is a pity to claim that this work is not recommended for the publication on Biomimetics, due to insufficient understanding toward this subject, inappropriate section arrangement and even some scientific mistakes. Detailed comments are shown below.
1. The Introduction part starts with the discussion with biomaterials, but the authors do not provide a clear clarification on the relationship between “biomaterial” and “nanozyme”, making the logic of this part rather confused. Actually, it is believed that such detailed discussion and classification on “biomaterial” is rather unnecessary, as it is beyond the main scope of this review.
2. The arrangement of the review is rational, including the four main parts of introduction, structure and properties, classification and application. However, each part lacks scientific and in-depth discussion. For instance, it is known that one of the main challenges for nanozyme research is their much weaker enzymatic activity as compared to the natural counterparts. Several methodologies have been proposed to overcome this obstacle, including surface modification, enhancing atomic utilization efficiency with single-atom catalysts, resembling the microenvironment of natural active sites and so on. However, these points are not properly summarized in part Structure and property.
3. Following comment 2, the classification of nanozymes based on their functional metal element like in Table 1 is acceptable. However, the presenting form of this table is a “nightmare” for readers to correlate the specific medical application with the exact material, or to search for the exact literature. Worthy of mentioning, this table is not even cited in the manuscript.
4. In the Application part, the authors intend to summarize the “biomedical applications” of different nanozymes. However, this point is not highlighted in the title, which might cause a misunderstanding as “applications in all fields”. Also, if biomedical application is focused, how could Environmental Treatment (Part 4.5) be included in this part? Such arrangement is apparently irrational.
5. Some typical work should be introduced as figures, instead of the widely-known enzymatic mechanism such as peroxidase-like (POD-like) activity in figure 3 and oxidase-like activity in figure 4, not to mention the non-scientific illustration on POD-like activity with duplicate terms of H2O2, and the lack of necessary reducing substrates for the enzymatic reaction.
The English writing of this manuscript needs extensive editing with less grammar mistake and proper sentence segmentation for better understanding.
Author Response
The submitted manuscript makes a brief review with the topic of “nanozyme”, one emerging field of the research frontier. Based on a careful consideration, it is a pity to claim that this work is not recommended for the publication on Biomimetics, due to insufficient understanding toward this subject, inappropriate section arrangement and even some scientific mistakes. Detailed comments are shown below.
- The Introduction part starts with the discussion with biomaterials, but the authors do not provide a clear clarification on the relationship between “biomaterial” and “nanozyme”, making the logic of this part rather confused. Actually, it is believed that such detailed discussion and classification on “biomaterial” is rather unnecessary, as it is beyond the main scope of this review.
Ans: The introduction part is modified as per the reviewer’s suggestion
- The arrangement of the review is rational, including the four main parts of introduction, structure and properties, classification and application. However, each part lacks scientific and in-depth discussion. For instance, it is known that one of the main challenges for nanozyme research is their much weaker enzymatic activity as compared to the natural counterparts. Several methodologies have been proposed to overcome this obstacle, including surface modification, enhancing atomic utilization efficiency with single-atom catalysts, resembling the microenvironment of natural active sites and so on. However, these points are not properly summarized in part Structure and property.
Ans: Structure and property section has been restructured.
- Following comment 2, the classification of nanozymes based on their functional metal element like in Table 1 is acceptable. However, the presenting form of this table is a “nightmare” for readers to correlate the specific medical application with the exact material, or to search for the exact literature. Worthy of mentioning, this table is not even cited in the manuscript.
Ans: The table has been cited in the manuscript.
- In the Application part, the authors intend to summarize the “biomedical applications” of different nanozymes. However, this point is not highlighted in the title, which might cause a misunderstanding as “applications in all fields”. Also, if biomedical application is focused, how could Environmental Treatment (Part 4.5) be included in this part? Such arrangement is apparently irrational.
Ans: Rearranged
- Some typical work should be introduced as figures, instead of the widely-known enzymatic mechanism such as peroxidase-like (POD-like) activity in figure 3 and oxidase-like activity in figure 4, not to mention the non-scientific illustration on POD-like activity with duplicate terms of H2O2, and the lack of necessary reducing substrates for the enzymatic reaction.
Ans: Done
Comments on the Quality of English Language
The English writing of this manuscript needs extensive editing with less grammar mistake and proper sentence segmentation for better understanding.
Ans: Whole manuscript was revised.

Round 2
Reviewer 1 Report
No further comments
Author Response
Thank you for the refinement in the previous round and thanks again for the consideration.
Reviewer 2 Report
The text requires further revision as the corrections have not been fully completed. References to figures were not included in the text. Although the conclusion briefly mentions the difficulties of working with nanozymes that need to be resolved before their practical implementation, it would benefit from a more detailed description. There is a substantial scientific literature available on this topic, providing sufficient information.
Author Response
As per the reviewer suggestion, the manuscript has been revised completely and references to figures in text has been included.

Reviewer 4 Report
The manuscript could be published on Biomimetics in the present form.
Author Response
The authors thank the reviewer for the consideration of manuscript for publication.
